# Parity-Adjusted Term Neonatal Growth Chart Modifies Neonatal Morbidity and Mortality Risk Stratification

**DOI:** 10.3390/jcm11113097

**Published:** 2022-05-30

**Authors:** Roie Kofman, Rivka Farkash, Misgav Rottenstreich, Arnon Samueloff, Netanel Wasserteil, Yair Kasirer, Sorina Grisaru Granovsky

**Affiliations:** 1Department of Internal Medicine, Hadassah Medical Center, Faculty of Medicine, Hebrew University of Jerusalem, POB 12000, Jerusalem 91120, Israel; roie.kofman@mail.huji.ac.il; 2Department of Obstetrics & Gynecology, Shaare Zedek Medical Center, Faculty of Medicine, Hebrew University of Jerusalem, Jerusalem 91120, Israel; rivka_f@szmc.org.il (R.F.); smuelof@cc.huji.ac.il (A.S.); sorina@szmc.org.il (S.G.G.); 3Department of Pediatrics, Shaare Zedek Medical Center, Faculty of Medicine, Hebrew University of Jerusalem, Jerusalem 91120, Israel; netanel.wasserteil@gmail.com (N.W.); yairkasir@szmc.org.il (Y.K.)

**Keywords:** birth weight charts, small for gestational age, large for gestational age, neonatal outcomes

## Abstract

Objective: To investigate the impact of parity-customized versus population-based birth weight charts on the identification of neonatal risk for adverse outcomes in small (SGA) or large for gestational age (LGA) infants compared to appropriate for gestational age (AGA) infants. Study design: Observational, retrospective, cohort study based on electronic medical birth records at a single center between 2006 and 2017. Neonates were categorized by birth weight (BW) as SGA, LGA, or AGA, with the 10th and 90th centiles as boundaries for AGA in a standard population-based model adjusted for gestational age and gender only (POP) and a customized model adjusted for gestational age, gender, and parity (CUST). Neonates defined as SGA or LGA by one standard and not overlapping the other, are SGA/LGA CUST/POP ONLY. Analyses used a reference group of BW between the 25th and 75th centile for the population. Results: Overall 132,815 singleton, live, term neonates born to mothers with uncomplicated pregnancies were included. The customized model identified 53% more neonates as SGA-CUST ONLY who had significantly higher rates of morbidity and mortality compared to the reference group (OR = 1.33 95% CI [1.16–1.53]; *p* < 0.0001). Neonates defined as LGA by the customized model (LGA-CUST) and AGA by the population-based model LGA-CUST ONLY had a significantly higher risk for morbidity compared to the reference (OR = 1.36 95% CI [1.09–1.71]; *p* = 0.007) or the LGA POP group. Neonatal mortality only occurred in the SGA and AGA groups. Conclusions: The application of a parity-customized only birth weight chart in a population of singleton, term neonates is a simple platform to better identify birth weight related neonatal risk for morbidity and mortality.

## 1. Introduction

Birth weight is a major predictor of neonatal morbidity and mortality and is used for tailoring the delivery decision-making process and future prenatal care. As a result, the definition of neonatal weight as appropriate or inappropriate for gestational age has major clinical and financial implications. Neonatal birth weight below the 10th percentile, the small for gestational age (SGA), and above the 90th percentile, the large for gestational age (LGA), being associated with increased perinatal morbidities and mortality [1,2,3], even after a term pregnancy [4,5,6,7,8,9,10,11,12].

Previous studies have suggested that customized birthweight curves may provide a better predictive tool for the adverse neonatal outcome than the traditional population-based gender-adjusted curve [13,14,15,16], although others found no significant difference in prediction of morbidity and mortality [17,18,19]. The design and data quality of many of these studies have recently been criticized [20], and more evidence is needed to resolve this debate. Furthermore, the vast majority of previous studies have focused on SGA neonates, leaving the LGA population understudied.

Among the many factors that influence neonatal birth weight, parity is a substantial determinant [21]. The maternal population attended by our center is characterized by high motivation for large families and parity. Thus, based on this natural model population, we aimed to investigate the impact of parity-customized and population-based birth weight charts on the neonatal adverse outcomes in small (SGA) or large for gestational age (LGA) infants compared to appropriate for gestational age (AGA) infants.

## 2. Materials and Methods

### 2.1. Study Population

Data were retrieved from the electronic medical charts (EMC) of all women who gave birth at the Shaare Zedek Medical Center (SZMC) in 2006–2017. All mothers and neonates benefit from antenatal and birth health care coverage of the Israel National Health Insurance Plan. Similar to others, for the purpose of the present study in order to better isolate the effect of parity on BW curves, we included uncomplicated, singleton pregnancies resulting in term births, 37–42 weeks’ gestation. The exclusion criteria for the study were: any maternal record of background of chronic illness or pregnancy complications (i.e., all categories of diabetes, hypertension, hypothyroidism, inflammatory bowel disease, and autoimmune diseases etc.,) [22,23,24,25]. All modes of delivery were included.

### 2.2. Study Design

This is an observational, retrospective, cohort study. Each delivery was categorized into one of ten parity groups, from parity 1 to ≥10, and the mean birth weight was calculated for each parity group and gestational age.

The cohort was used to generate new customized birth weight centiles, adjusted for fetal gender, week of gestation, and maternal parity. Each neonate was then classified as SGA, AGA, or LGA, using the 10th and 90th percentiles for BW as cutoffs, based on two models:

The standard population-based growth chart was used in Israel, adjusted for gestational age and neonatal gender only [26] (POP).

The new growth chart we generated was customized for gestational age, neonatal gender, and maternal parity (CUST).

Data regarding adverse neonatal outcomes were extracted for the cohort population. Data regarding the following adverse outcomes were included: meconium aspiration (ICD9 770.12), hypoglycemia (ICD9 775.6), 5′ Apgar < 7, admission to the neonatal intensive care unit (NICU) for over 72 h, shoulder dystocia (ICD9 660.4), and neonatal death, defined as death within 28 days of birth.

For further analysis, the study population was then reclassified to create the “groups of disagreement” between the two models.

Finally, to investigate the risk associated with each of the “groups of disagreement”: SGA-CUSTONLY, SGA-POP ONLY, LGA-CUST ONLY, and LGA-POP ONLY, a multivariable model was generated for a composite adverse outcome and each adverse outcome individually. Neonates in the 25–75th centiles of weight served as a reference population for the analyses. All models in the study were adjusted for the following variables: parity, maternal age, conception by in vitro fertilization (IVF), and use of epidural or oxytocin during labor. Odds ratios (ORs) for composite adverse outcomes were calculated with 95% confidence intervals (CIs). A separate model, adjusted for the same variables and using the same reference population, was used to investigate the risk of shoulder dystocia among LGA neonates.

### 2.3. Statistical Methods

Customized birth weight percentiles were calculated separately for each gender in all term singleton deliveries according to gestational age and parity. Mean birth weights of all parity groups were compared using one-way ANOVA. Comparisons between the distribution of national population percentiles and customized percentiles were performed using the Chi-square test.

For the purpose of the study, we defined the following study groups: (1.1) SGA by population percentile only; (1.2) SGA by customized percentile only; (1.3) SGA by both percentiles; (2.1) LGA by population percentile only; (2.2) LGA by customized percentile only; (2.3) LGA by both percentiles and (3) non-SGA and non-LGA. Univariate analysis (Chi-square test) was used to compare the rates of any adverse neonatal outcome for (1.1), (1.2), and (1.3) vs. (3) and for (2.1), (2.2), and (2.3) vs. (3).

Multivariable logistic models were used to assess the independent role of extreme neonatal birth weight according to all definitions of neonatal adverse outcomes, using neonates in the 25–75th birth weight percentiles as a reference group.

All tests were two-sided. *p* < 0.05 was considered statistically significant. Analyses were carried out using SPSS Statistics for Windows, Version 25.0. (IBM Corp., Armonk, NY, USA).

### 2.4. Ethics

This study was approved by the SZMC research ethics review board (Helsinki committee authorization number 0154-17-s2).

## 3. Results

During the study period, 168,979 births were registered at SZMC, of which 165,633 (98%) were singleton and 157,969 (93.5%) were singleton term births (37–42 weeks’ gestation).

We excluded 24,813 deliveries (14.7%) due to chronic background maternal medical conditions and pregnancy complications and 249 (0.001%) were excluded due to intrauterine fetal demise, yielding 132,907 deliveries, 92 were excluded from the analysis due to errors in the documentation of data. As a result, 132,815 deliveries (78.59% of all deliveries) by 69,955 women met the study inclusion criteria (Figure 1). Overall, 37,709 (51%) of the mothers contributed one delivery to the study and 34,246 (49%) contributed ≥2 deliveries. The population characteristics are shown in Table A1. The maternal cohort included 24.3% primiparas, 59.3% multiparas, 14.9% grand-multiparas (parity 5–9) and 1.5% grand-grand multiparas (parity ≥ 10). See Appendix A for full data (Table A1).

Calculation of mean birth weight for each maternal parity group demonstrated a progressive increase in birth weight from a mean of 3220 g ± 405 for primiparas up to a mean birth weight of 3498 ± 450 for parity group ≥ 10 (*p* < 0.001, Figure 2).

### 3.1. Birthweight

Birthweight data are shown in Table A2 and Table A3 and Figure 3. Overall, 8894/132,815 neonates (6.7%) were defined as SGA using the population-based chart (SGA POP), compared to 13,334 (10%) defined as SGA using the parity-adjusted model (SGA CUST). In the SGA CUST group, 4742/13,334 (35.5%) neonates were defined as SGA CUST ONLY; in the SGA POP group, 302/8894 (3.4%) were defined as SGA POP ONLY. A total of 8592/132,815 (6.4%) neonates were defined as SGA BOTH and 119,179/132,815 (89.7%) were defined as non-SGA.

There were 13,348 neonates (10%) defined as LGA using the population-based chart (LGA POP), while 13,155 (9.9%) were defined as LGA using the parity-adjusted model (LGA CUST). Within the LGA CUST group, 2068/13,155 (15.7%) neonates were defined as LGA CUST ONLY, while in the LGA POP group, 2261/13,348 (16.9%) neonates were defined as LGA POP ONLY. A total of 11,087/132,815 (8.3%) neonates were defined as LGA BOTH and 117,399/132,815 (88.3%) were defined as NON-LGA.

### 3.2. Overall Neonatal Risk of Morbidity and Mortality

Overall, 5656 neonates (4.25%) experienced any of the previously defined adverse outcomes. The most common adverse outcome was hypoglycemia (*n* = 3640; 2.7%) and the least common was neonatal death (*n* = 12; 0.009%). Notably, all cases of neonatal death occurred within the AGA and SGA groups. Detailed data regarding the distribution of outcomes among the birth weight groups are in the Appendix A (Table A2 and Table A3).

### 3.3. Adverse Neonatal Outcome Analysis for Each Birth Weight Definition Groups

#### Univariable Models

**SGA:** Neonates in the SGA-BOTH group had the highest rate of adverse outcomes among the SGA groups, which were experienced by 7.9% of neonates (682/8592; *p* < 0.001). The SGA-POP ONLY group experienced a slightly higher rate of adverse outcomes than the SGA-CUST ONLY group (5.6% [17/302]; *p* < 0.001 and 5.5% [261/4742]; *p* < 0.001, respectively). The NON-SGA group had the lowest adverse outcome rate of 4.1% (4927/119,179; among the SGA groups. *p* < 0.0001). SGA data are shown in Table A2.

**LGA:** Neonates in the LGA-BOTH group had the highest rate of adverse outcomes among the LGA groups, which were experienced by 7.7% (856/11,087; *p* < 0.001). The LGA-POP ONLY group had a higher rate of adverse outcomes compared to the LGA-CUST ONLY group (5.9% [134/2261]; *p* < 0.001 and 4.9% [101/2068]; *p* < 0.001, respectively). The NON-LGA group had the lowest rate of adverse outcomes (4.4% [5222/117,399]; *p* < 0.0001). The highest rate of shoulder dystocia among the LGA groups occurred in the LGA-BOTH group (48% [204/426, *p* < 0.001). The differences in the rate of meconium aspiration and NICU > 72 h did not reach statistical significance. LGA data are shown in Table A3.

### 3.4. Multivariable Models

The risk of neonatal composite adverse outcomes was increased in all SGA groups. The SGA-CUST ONLY group showed increased risk for composite adverse outcomes compared to the reference population (OR = 1.33 95% CI [1.16–1.53]; *p* < 0.0001), but lower than the SGA-BOTH group. Confidence intervals overlapped between the SGA-POP ONLY and SGA-CUST ONLY groups (Table A4). With respect to individual outcomes, only the risk for hypoglycemia was significantly increased for both SGA groups (Figure 4). Confidence intervals for all other calculated ORs partially overlapped for the CUST and POP models.

In the LGA population, the risk for any adverse outcome was increased significantly for LGA BOTH and LGA-CUST ONLY (OR = 1.62 95% CI [1.48–1.77]; *p* < 0.0001; and OR = 1.36 95% CI [1.09–1.71]; *p* = 0.007, respectively), compared to the reference population. However, neonates defined as LGA-POP ONLY were not at significantly increased risk for the neonatal composite adverse outcome (OR= 1.19 95% CI [0.97–1.45]; *p* = 0.092) (Table A4). The risks of hypoglycemia and shoulder dystocia were significantly increased for all LGA groups (Figure 5).

The risk of shoulder dystocia was highest in the LGA BOTH group (OR = 16.6 95% CI [12.63–21.82], *p* < 0.0001) and was also elevated almost seven-fold in the LGA-CUST ONLY and LGA-POP ONLY groups (Table A5, Figure 5).

## 4. Discussion

Our study based on a population, constituting mainly Orthodox Jewish and Muslim women with exceptionally high parity, has the potential to shed light on the neonatal birth weight outcome with a focus on parity; i.e., the effect of customization for parity as a sole and simple variant, using a large maternal cohort from a single center. Indeed, we observed a gradual rise in birth weight as parity increases, up to parity ≥10; however, the largest increase occurred from parity 1 to 2.

These results agree with some of the studies exploring this association [13,21,27,28,29,30]. Hinkle et al. [31] identified a similar nonlinear increase in birth weight; also, after adjustment for other confounders such as maternal height, body mass index (BMI), and smoking. Interestingly, Hinkle described a trend of birthweight stabilization between parity 4–7. This trend was not observed in our cohort, but rather little sustained increments. This may be consistent with the placentation in previous pregnancies theory [32,33,34]. Of note, most previous studies included a relatively small number of grand multiparas; one of the unique features of our study population is the very high parity, with a median parity of 3 deliveries per mother, while 16.4% of deliveries were over parity 6.

Specially, the present study classified the neonatal population into birthweight categories using both population-based (SGA-POP) and customized (SGA-CUST) models, and risks were estimated and compared accordingly. The neonatal risks identified by this classification are mainly related to the SGA neonates, the most challenging neonatal population. Using the parity-customized definition, an additional 4742 neonates were defined as SGA, a 53% increase from the number characterized as SGA based on the population-based definition. This large difference can be explained by the higher parity of our study population relative to the general Israeli population used to generate the population-based charts. Neonates that were defined as SGA according to either definition had higher overall rates of adverse outcomes than non-SGA newborns. Notably, all cases of neonatal death occurred in the SGA and AGA groups, either population- or parity-adjusted. This may be due to exclusion of maternal morbidities, mainly diabetes, in our cohort. Of note, the overall mortality rate for our cohort was exceptionally low (12 out of 132,815 cases). The SGA-CUST ONLY group experienced a higher number of individual cases for each adverse neonatal outcome, but a lower adverse outcome rate hypothetically to be attributed to increased sensitivity at the expense of lower specificity resulting from the customized SGA definition. The higher morbidity in the SGA group is even more remarkable for a maternal study population with no chronic background disease or pregnancy complications; thus, SGA represents an intrinsic neonatal risk for this group [35].

The SGA-CUST ONLY group showed a 33% increase in the risk for composite neonatal adverse outcomes compared to normal birth weight neonates. The SGA-POP ONLY group, which included neonates defined as SGA by the population-based chart but not by the customized model, also had increased risk of the composite adverse outcome, but included a relatively small group, with just 302 neonates experiencing 17 cases of adverse outcomes. In the present study we excluded maternal chronic background diseases or pregnancy complications, which are associated with SGA; thus, in a general population the increased risk for adverse outcomes seen for the SGA neonates, adjusted for parity, might become even more significant. These results, strengthen the notion that the use of the customized chart might be beneficial in the population of SGA neonates.

Studies [13,36] using fully customized models (including parity, maternal height, BMI, ethnicity, and other variables) that included parity had similar results and conclusions regarding the benefit of customization. A meta-analysis by Chiossi et al. [37]. on population versus customized growth charts use concluded that the customized models resulted in a higher overall pooled ORs for adverse neonatal outcomes, yet the authors refrained from concluding that the customized models were superior due to overlapping 95% CIs. A French study by Ego et al. [19]. concluded that customization for parity does not improve the detection of high-risk low-birthweight neonates. A possible explanation for the discrepancy with our study might be the use of different inclusion criteria. The Ego study included preterm births, which were excluded in our study, and more than half of were primiparas compared to only 24% of our population.

Regarding the LGA population, both models defined a comparable number of neonates, about 10% of the cohort, as LGA, with the population-based chart (LGA-POP) including just 1.5% more neonates in this group. About 15% of the members of each group did not overlap with the other. Unlike the large difference in SGA populations, this smaller difference in LGA populations might stem from the fact, that the most significant impact of parity on birth weight manifests in parity 1 to 2, a neonate population more prone to low birth weight. This diminishing impact of parity in the higher birth weights may lead to a less pronounced difference using the customized model. Importantly, the neonates in the LGA-CUST ONLY subgroup had a 36% higher risk of composite adverse outcomes compared to those in the reference population, while the difference in risk for LGA-POP ONLY neonates did not reach statistical significance. This may indicate the ability of the parity-customized model to separate between neonates who are constitutionally large and those with an underlying pathological condition, a possible advantage of the parity customized model in the LGA population.

Indeed, the impact of customized charts for the term LGA is disputable. Chiossi et al. in a meta-analysis [37] concluded that apart from shoulder dystocia, neither the population nor the customized models discriminate large neonates with significantly increased risk of adverse outcomes. The results in our study might again stem from differences in the customization models, study variables, and from vast differences in characteristics of the study population.

The main strengths of this study include the large cohort of women with high parity and their equal access to health insurance and prenatal care. This large sample with well documented validated records allowed us to isolate the parity and uncomplicated pregnancies, thus eliminating additional risks associated with maternal or fetal compromise. These background maternal diseases challenge any systematic review and meta-analyses of customized growth curves [37]. Other strengths include the exclusion of preterm births and maternal diseases, factors that may be associated with parity and hence act as possible confounders and may have effects on neonatal outcomes that are not specifically linked to birth weight. Limitations of the study include the lack of adjustment for maternal factors that may influence birth weight such as maternal BMI, height, and smoking, an inherent constraint due to our data collection methods. However, in the study by Hinkel et al. [24], which analyzed the influence of many possible confounders, the authors concluded that none of these confounders explained this association between parity and birthweight. Fortunately, for the term neonates, neonatal death in our health system is a rare event. Thus, we considered other outcomes, frequently used in similar studies; however, with some limitations. We used the outcome of prolonged NICU admission (>72 h); this outcome may be inclusive of other adverse neonatal outcomes. However, the evaluation of this outcome, was considered in fact exactly because of its potential to be inclusive; thus, we made sure by the use of this outcome to include adverse outcomes that might have been missed in the coding of the records diagnoses; even more it was evaluated and reported by us as a separate outcome. All adverse outcomes were used as recorded and not by their changing definition; i.e., hypoglycemia. Indeed, the definitions may have changed however for each certain maternal parity the diagnosis was reported in a similar mode. Our study is also limited by the generalizability of the results to other different populations.

## 5. Conclusions

In conclusion, we have explored the impact of a customized birth weight chart based on parity alone, in a population of singleton, term neonates born to healthy mothers. We found that this simple customized definition especially for the SGA neonates, may enable a more precise identification of those at higher risk for adverse outcomes in parous mothers. These results support the importance of maternal parity as a meaningful variable for customization models, especially for future machine-learning algorithms.

## Figures and Tables

**Figure 1 jcm-11-03097-f001:**
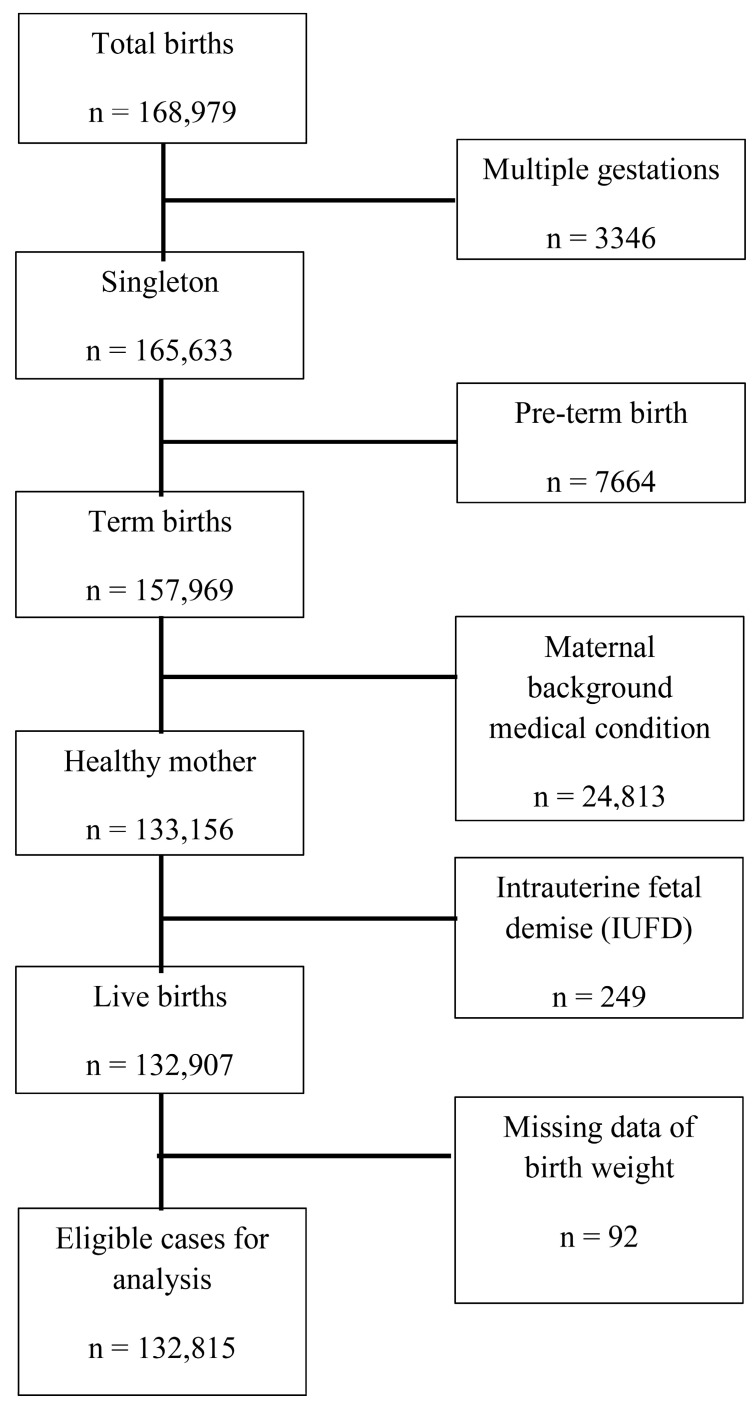
Study population flowchart.

**Figure 2 jcm-11-03097-f002:**
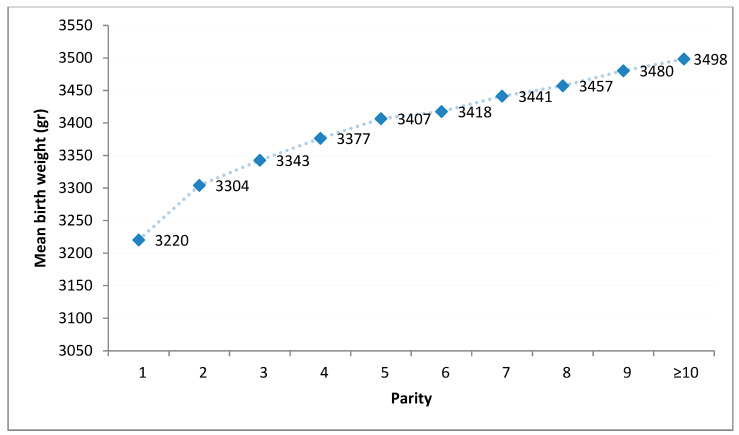
Birth weight by parity for the entire study cohort (mean birth weight).

**Figure 3 jcm-11-03097-f003:**
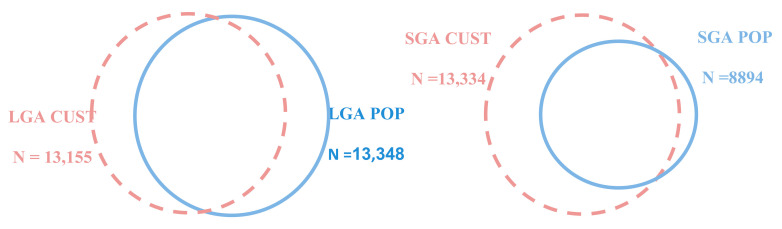
The relationship between the SGA and LGA study group groups (Venn diagram). LGA, large for gestational age; CUST, customized model adjusted for gestational age, gender, and parity; SGA, adverse outcomes in small; POP, population-based model adjusted for gestational age and gender only.

**Figure 4 jcm-11-03097-f004:**
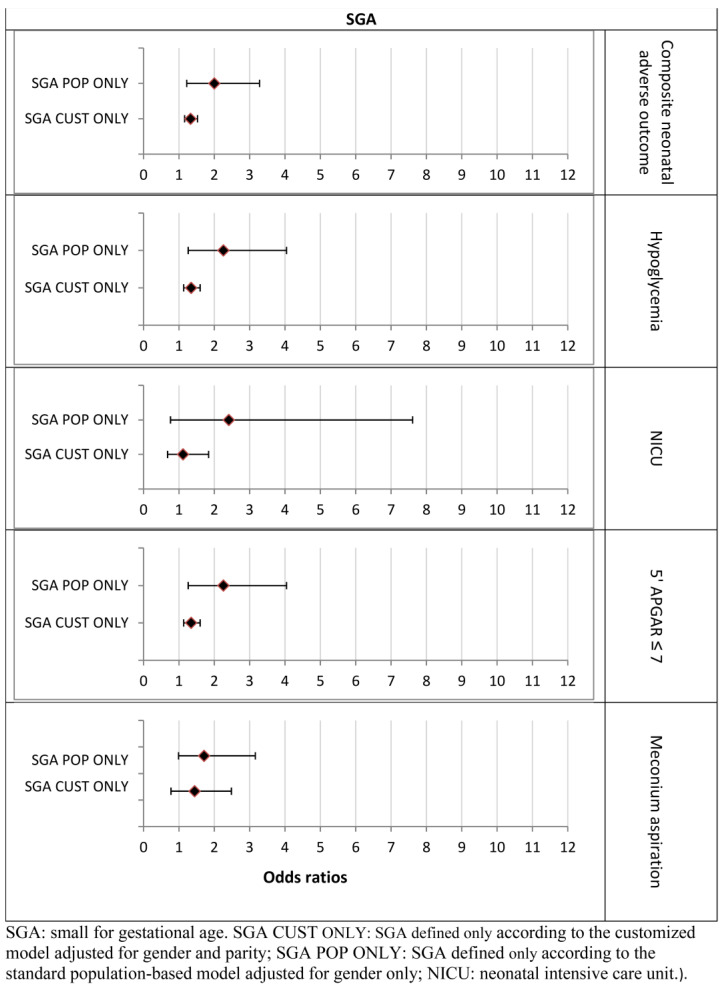
The association between SGA group categorization and adverse neonatal outcomes (Odds ratios, 95% CI). The customized models are adjusted for maternal age.

**Figure 5 jcm-11-03097-f005:**
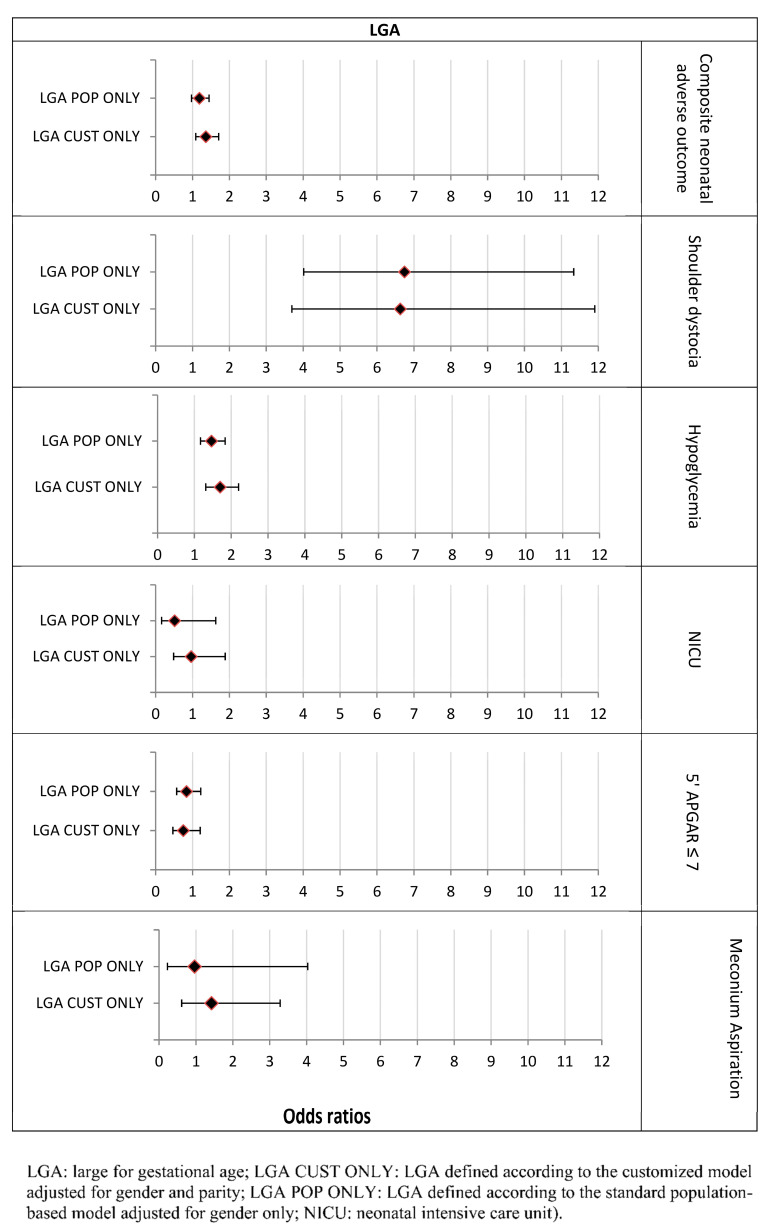
The association between LGA group categorization and adverse neonatal outcomes (Odds ratios, 95% CI). The customized models are adjusted for maternal age.

## Data Availability

Not applicable.

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
