# Peer review of "Parity-Adjusted Term Neonatal Growth Chart Modifies Neonatal Morbidity and Mortality Risk Stratification"

_jcm, 2022, doi:10.3390/jcm11113097_

Round 1
Reviewer 1 Report
The authors used the data to develop a birth weight chart model adjusted for gestational age, gender,and parity, and evaluated its values for neonate morbidity and mortality risk in a characterized population. The results identify its meaning, and the conclusion is senses reasonably.
Some comments:
The classification may be more complex regarding the adverse neonatal outcomes, and a clear explanation is needed. Why choose these neonatal outcomes? Some outcomes are overlapped. For example, the outcome of NICU >72 h may contain meconium aspiration and hypoglycemia simultaneously. Also, the definitions of adverse neonatal outcomes may not be the same during a ten-year span, as some guidelines, standards (such as hypoglycemia), and clinical treatment plans were changed with time.
Table 4 and Table 5 are not found in the Appendix.
Reviewer 2 Report
This study compared the risk of adverse perinatal outcomes between two birth bodyweight statuses using electronic medical data from a single center. The feature of this study is that women have more children in Israel, and more than 10% of the participants had a parity of more than 5 in this unique population. However, several critical issues in this study need to be clarified, and the English writing needs to be revised.
I have some suggestions for the authors and hope that my comments are constructive to them.
- The definition is very important. The current definition of SGA or LGA, if considering population level, is it reasonable to exclude related maternal chronic diseases? Is the relevant literature also defined in this way? Or do most of the researchers include women with preeclampsia and GDM? Please cite the literature to support your study definition.
- The authors considered parity a critical factor in assessing birth weight and predicting adverse perinatal outcomes. However, the number of births may highly correlate with the age of the mothers. How to determine whether the difference is related to parity or the maternal age at childbearing? Please provide possible corroboration or explanation.
- Line 86-100 and 108-113. These two paragraphs are highly repetitive, and please combine the details of the research method.
- Please provide high-resolution pictures to illustrate your study findings.
- Please list the birth weight ranges that define SGA and LGA, which can record in the supplementary information for readers' reference.
- Did mothers with GDM exclude in your study? Please clarify.
- Line 284, your reference # 34 is wrong; please confirm it and check your references carefully.
